# Competition Strategies for Location-Based Mobile Coupon Promotion

**Pengcheng Xia [1], Gang Li [1,\*], T.C.E. Cheng [2] and Ao Shen [1]**

[1] The Management School, Xi'an Jiaotong University, Xi'an 710049, China; xiapengcheng@stu.xjtu.edu.cn (P.X.); shenao_moira@163.com (A.S.)
[2] Department of Logistics and Maritime Studies, The Hong Kong Polytechnic University, Hong Kong 999077, China; edwin.cheng@polyu.edu.hk
[\*] Correspondence: glee@mail.xjtu.edu.cn

**Abstract:** Location-based service heightens consumers' shopping convenience. By utilizing spatial flexibility of consumers, retailers can target consumers via location-based mobile coupons (LBMCs) to enhance market performance. Considering the strategies for LBMC promotion for two competing retailers, we find that under different market intensities, only no adoption and symmetric adoption of LBMC promotion are the possible equilibria for the competing retailers at a low marginal targeting cost. Then, we extend our model to consider vertical (quality) differentiation and analyze the implications of adopting LBMC promotion for a superior-quality firm (with higher product valuation) and an inferior-quality firm (with lower product valuation). Mixed strategies for LBMC promotion emerge when firms' products have different qualities. Our research findings provide useful guidance for managers and marketing practitioners to formulate strategies for targeted LBMC promotion.

**Keywords:** e-commerce; mobile coupons; location-based services; game theory

## 1. Introduction

Mobile technologies are changing the way firms do business [1,2]. Location-based services (LBSs), i.e., GPS-authorized smartphones that reveal the phone users' locations, allow marketers to target customers by their locations [3]. From the consumers' points of view, mobile devices with LBSs have two major advantages, namely time and space flexibility, over the conventional desktop computer [4]. In other words, consumers can seek and buy their desired products from mobile stores anytime and anywhere, both of which are independently and mutually constrained by traditional bricks-and-mortar channels [5]. Luo et al. [6] showed that consumers' purchase aspirations increased when they receive promotional offers close to the time and place of the promotional campaign. Hence, combining geographical and temporal targeting is a potent strategy for companies to engage customers at the right moment and in the right location, which can stimulate consumer responses [7] and increase purchases [6].

Recent advances in LBSs provide businesses with a new marketing channel for mobile promotion [8]. Mobile promotion enables direct communication with target consumers with individualized information anytime and anyplace [9]. As an emerging approach, mobile coupons marketers are playing an increasingly important role in mobile promotion activities. Mobile coupons are digital coupons that companies send to consumers as promotional offers, commonly in the form of short message services (SMSs), applications (APPs), mobile websites, or emails [10,11]. The overwhelming majority of early mobile promotion forms are SMSs [12,13]. Consumers can save these coupons in their mobile devices and redeem them later at the point of sales. In 2019, 65% of respondents said receiving mobile coupons they can redeem in-store is important when shopping in physical stores. Additionally, 69% said receiving a personalized offer on their phone that they can use in-store would make them more likely to visit a physical retail location [14].

Apparently, coupons facilitate price discrimination and engender market segmentation that firms deploy to compete for price-sensitive consumers, be they paper or electronic in

nature. However, the traditional coupons cannot target consumers' locations and usage behaviors (firms can use demographic information [15] and panel data on household purchase behavior [16] to target different consumers); traditional coupons have higher printing and distribution costs as compared with e-coupons (however, non-targeting e-coupons merely reduce cost but cannot improve targeting efficiency). As a form of the third degree of price discrimination [17], mobile coupons endow firms with the capability to target different prices to different consumers based on consumer location information and behavioral information at a lower distribution cost and a higher targeting efficiency.

In this study we focus on location-based mobile coupon (LBMC) promotion, which constitutes a promising customer acquisition tool because offers in close vicinity of points of sale have potentially high relevance for consumers, i.e., geo-fencing. Instead of putting up a geo-fence around a store's own location to trigger mobile offers or discounts, a retailer or brand will geo-fence the locations outside of the firm's primary turf or even the locations of nearby rivals to pull in consumers, which is known as "geo-conquesting", i.e., "*mobile coupons target consumers with geo-fencing*" [18].

Theoretical studies of competitive bilateral targeted promotion and mobile coupons are surprisingly limited in prior research. Chen et al. [19] studied competitive mobile targeting based on consumers' real-time locations and concluded that a firm's profit could be higher under mobile targeting than under uniform pricing. This conclusion is, to some extent, in line with our finding that a firm that adopts LBMC promotion obtains a higher profit. Endeavoring to fill these research gaps, we address several issues concerning strategies for LBMC promotion in the context of two competing retailers. Specifically, we seek to address the following questions:

(1)  How should the competing retailers adopt targeted LBMC promotion?
(2)  How is the equilibrium strategy for LBMC promotion influenced by the competition intensity of the retailers and by the retailers' capabilities to target customers?

We answer these questions by developing a model of two competing retailers in which each retailer can send mobile coupons via mobile targeting technologies to areas in the residual market where the consumers have non-positive utilities to compensate for the disutility of spatial cost and attract the consumers beyond the retailer's market. We focus on analyzing a duopoly model in which the two competing retailers, first, choose whether to target consumers via mobile coupons, and then compete for consumers by simultaneously choosing their retail prices and the face values of their mobile coupons. Thus, our study entails endogenous mobile coupon decisions and pricing strategies in the context of LBMC promotion. We also extend the model to study the impact of product quality differentiation on the strategies for LBMC promotion.

We make three contributions to the literature on targeted LBMC promotion. First, we demonstrate that mobile coupons are an effective marketing tool to induce more location-conscious consumers by compensating consumers who are far away from the retailer's market. Second, we derive the conditions under which retailers should adopt LBMC promotion under different market competition intensities and at different targeting costs. We find that under different competition intensities, only no adoption and symmetric adoption of LBMC promotion are the possible equilibria for the competing retailers. Finally, we analyze the retailers' strategies for LBMC promotion competing under asymmetric quality. Interestingly, unlike the case of symmetric quality, we find that mixed strategies for LBMC promotion emerge despite the fact that the retailer with superior quality always makes a higher or the same profit as compared with that of the retailer with inferior quality.

We organize the remainder of this paper as follows: In Section 2, we present a brief review of the related literature; in Section 3, we introduce the oligopoly and duopoly models as the basic tools for analysis; in Section 4, we derive the equilibrium strategies; in Section 5, we extend the model to consider quality differentiation; in Section 6, we conclude the paper, discuss the managerial implications of the research results, and suggest topics for future research.

## 2. Literature Review

Our research is closely related to the literature on mobile promotion, targeted marketing, and mobile coupons. In the following sections we provide a brief review of the pertinent literature on each of these topics.

### 2.1. Mobile Promotion

Mobile marketing has gained increasing popularity in recent years, which refers to two-way or multi-way communication and promotion of an offer between a firm and its customers using a mobile medium, device, or technology [1]. The traditional model of retailing is based on consumers entering the retailing environment, making location the primary source of competitive advantage [20]. However, given that the extent of users' geographical mobility has a positive effect on their mobile Internet activities [21], firms can precisely target customers in any place, including rivals' service areas. Therefore, retailers can now enter the consumer's environment and, because the mobile device stays with the consumer, the retailer can be anywhere and anytime [2]. for more details about drivers of mobile adoption and the influence of mobile marketing on customer decision making, which Shankar et al. [2] extended to focus on the retailing environment.

The majority of previous studies on mobile promotion have been location irrelevant. Studying permission-based advertising via SMS on mobile phones, Barwise and Strong [12] suggested that the mobile channel had the potential to benefit both advertisers and consumers. Dickinger and Kleijnen [13] argued that firms should not flood consumers with mobile coupons; they should consider the usability of mobile coupons in deciding the right offer, which demonstrated the necessity for precision location and timing information about customers. Some studies have examined location-relevant promotion. For example, Hui et al. [3] used RFID tracking in conjunction with an entrance and exit survey to investigate the effect of in-store travel distance on unplanned spending. Fang et al. [22] found that location-based mobile promotion had a significantly stronger impact on contemporaneous purchases and delayed purchases. Baardman et al. [23] developed a new personalized demand model with customer trend for targeted promotions to the right customers, which helped to find important customers and target them to generate additional sales.

However, to the best of our knowledge, previous research has not addressed the question of how to compete for location-targeted customers by taking advantage of LBSs and mobile coupons, although an increasing nuber of firms are adopting this promotion strategy. Moreover, there has been no empirical and theoretical analyses of the ideal competitive response to a competitor's adoption of mobile promotion [8]. We endeavor to fill this gap by providing a theoretical analysis of the strategies of two competing retailers engaged in targeted LBMC promotion [24]. We find that both retailers have incentives to adopt LBMC promotion to generate higher profits. The non-adopter is not affected when the competition intensity is not high; but this is not the case when the competition intensity is relatively high. There is a unique subgame-perfect Nash equilibrium (SPNE) at which either both retailers adopt LBMC promotion or not. We also find that the marginal cost of offering mobile coupons is a crucial determinant for firms to adopt LBMC promotion under relevant competition intensities.

### 2.2. Targeted Marketing

Targeting with consumer attributes is widespread in the marketing literature. Previous research has examined location-specific pricing based on panel data on household purchase behavior [16] and price discrimination based on customers' recognitions of their past purchase behaviors [25–28]. Hartmann [29] integrated social interactions into tracking consumption behavior and evaluated group-specific characteristics for targeted decision making. But the information acquired from internal sources such as firms' transaction databases or from external sources such as credit reports may cause imperfect targeting of prices to customers [30]. Ensuring that the media are efficiently directed to the "right people" is an important decision that a marketer makes. Targeted advertising allows the

firm to eliminate "wasted" investment in consumers whose preferences do not match its product's attributes [31]. In addition, online and offline channels may be complements with regard to targeted advertising [32]. However, Fong et al. [33] showed that targeted promotions increase promoted product sales and purchases of similar products; they can crowd out purchases of dissimilar products (i.e., e-books from nontargeted genres) by decreasing search activities of nontargeted goods on the same platform. In addition, Luo et al. [34] examined whether and how inducing online shopping complements or cannibalizes a firm's offline sales. Their findings alert managers to the dangers of improper targeting and investment in information technology and the importance of consumer heterogeneity for omnichannel commerce across online and offline channels.

Location plays a key role in affecting consumers' purchasing behaviors [35], There is emerging research on location-based targeting. For example, Luo et al. [6] found a negative sales-lead time relationship of proximal mobile users and an inverted-U, curvilinear relationship of non-proximal mobile users through temporal and geographical targeting. Interestingly, Fong et al. [8] and Zhang et al. [36] quantified the effectiveness of competitive locational targeting based on a randomized field experiment, which merely focused on the effects of unilateral promotions. In their subsequent research, they showed that competitive price discrimination improved the performance of behavioral targeting but led to a slowdown in location-based targeting [17,37]. Wang and Li [38] examined the economic effects of consumer mobility on pricing and advertising strategies by incorporating consumers' Lévy-walking behaviors into advertising economics models. However, limited analytical research has been devoted to studying the strategy of location-based targeting in a competitive market, which frustrates a firm's implementation of location-based targeting promotion in the right way.

In this paper, we focus on location-based targeting, whereby firms push mobile coupons to areas in the residual market where consumers have non-positive utilities to compensate for the disutility of their temporal and spatial costs. Interestingly, consumers in the non-targeted segment are charged higher prices while those in the targeted segment are compensated for being in faraway locations. As a result, the total sale increases are in the form of the third degree of price discrimination such that different consumers are targeted with different prices based on their location information.

### 2.3. Mobile Coupons

There is a large body of the literature on coupons that deals with a multitude of issues, including coupon redemption for marketing research [39], airline coupon strategies [40], coupon redemption rates [41,42], profitability of coupons [43], competitive couponing [44,45], coupon face values [46,47], short- or long-duration coupons [46], and differences between coupons and rebates [48]. Multifarious coupons have also been analyzed, i.e., cents-off coupons [15], direct mail coupons [49,50], package coupons [51,52], cross-ruff coupons [53], and front-loaded versus rear-loaded coupons [54].

Mobile coupons serve as a complementary channel to send promotional offers to consumers [55]. The geographical and temporal features of mobile coupons [22] significantly influence redemption under different conditions of location and time [56]. Two fundamental issues concerning mobile coupons remain to be addressed: How should consumers adjust their coupon-using behavior in response to locational targeting, which affects their expectations? How should rival firms respond to competitive locational targeting? We address these issues in this study.

Previous studies have found that the effectiveness of mobile coupons depends on contextual variables and coupon characteristics [11]. For instance, by personalizing the displays of coupons, customers can cut through the coupon clutters, which will motivate them to purchase [57]. In contrast with previous studies, we focus on analyzing firms' strategies for adopting LBMC promotion, whereby firms choose to compete for consumers located in the remote segment for which mobile coupons play the important role of price discrimination. As compared with mass coupons, we find that mobile coupons take advan-

tage of consumers' various locations, which allows firms to exercise price discrimination to boost consumption demand. Thus, mobile coupons are an effective marketing tool that can compensate consumers for the disutility of their temporal and spatial costs.

## 3. The Model

### 3.1. Assumptions and Notation

We summarize the definitions of all the notation used in our model in Table 1.

**Table 1.** Definitions of notation.

| Notation | |
|---|---|
| $C$ | Adopt LBMC promotion |
| $NC$ | Not adopt LBMC promotion |
| $\pi_i$ | Firm $i$'s total profit |
| $d_i$ | Firm $i$'s total demand |
| $p_i$ | Firm $i$'s price of the product |
| $m_i$ | Face value of the mobile coupon offered by firm $i$ |
| $V$ | Consumer's willingness to pay for the product |
| $t$ | Consumer's unit travel cost |
| $c$ | Marginal cost of using mobile coupons to target the market |
| $T_i$ | Firm $i$'s targeting size via mobile coupons |
| $x_{jk}$ | Consumer's indifferent point between buying firm 1's and firm 2's products under the respective strategy set, $j$ and $k$ is an element of the set {$C$, $NC$}. |
| $\overline{x}_{jk,i}$ | Consumer's indifferent point between purchasing firm $i$'s product or not without the existence of mobile coupons under the respective strategy set, and $j$ and $k$ is an element of the set {$C$, $NC$}. |
| $\widetilde{x}_{jk,i}$ | Consumer's indifferent point between purchasing firm $i$'s product or not with the existence of mobile coupons under the respective strategy set, and $j$ and $k$ is an element of the set {$C$, $NC$}. |

### 3.1.1. Firms

Consider a market in which there are two firms (retailers) $i \in \{1, 2\}$ situated at the two ends of a Hoteling $[0, 1]$ linear city that compete for consumers by selling to them a horizontally differentiated product. Without loss of generality, we assume that the marginal cost of the product for each retailer is constant and normalized to zero.

The firms both target consumers within a particular range via mobile coupons and the marginal cost for the firms to target the market $[0, 1]$ is $c$. The cost to target a particular segment is linearly related to its size [31]. In reality, the costs of mobile coupons are driven by the size of the virtual barrier a firm sets and the number of customers the firm is supposed to reach.

Firm $i$ chooses whether to adopt LBMC promotion first. Then, each firm simultaneously sets its retail price and the face value of its mobile coupons. (Moraga-Gonzalez and Petrakis [58] modeled coupon advertising in a competitive market by simultaneously choosing the price, rebate, and intensity of couponing.) We make this assumption because the price decision can be changed rapidly according to market feedback and/or other factors in the wake of electronic commerce with mobile technology. While, before the age of the Internet, pricing was a long-term decision where managers started with an annual promotion schedule and only when the time for executing the promotion decision arrived, did that they decide whether to promote or not [59]. Shaffer and Zhang [16] also assumed in their study that coupon targeting decisions were made subsequent to the decisions on prices and coupon face values. Rao [60] observed that, in practice, regular prices were often chosen first, followed by the choice of promotion depth in a competitive environment.

### 3.1.2. Consumers

Uniformly distributed along the unit line, the consumers each demand only one unit of the product, incurring a unit travel cost $t > 0$ for traveling to the store for shopping. All the

consumers have access to mobile phones, which can receive mobile coupons precisely. We also assume that all the consumers redeem their mobile coupons for tractability. Consumer $l$'s utility from buying the product from firm $i$ without mobile coupons is in Equation (1):

$$U_l = \begin{cases} V - p_1 - tx_l \\ V - p_2 - t(1 - x_l) \end{cases}, without\ mobile\ coupons. \tag{1}$$

Consumer $l$'s utility from buying the product from firm $i$ with mobile coupons is in Equation (2):

$$U_l = \begin{cases} V - p_1 - tx_l + m_1 \\ V - p_2 - t(1 - x_l) + m_2 \end{cases}, with\ mobile\ coupons. \tag{2}$$

Consumer $l$ compares the net utilities of buying the product from firm $i$ with and without mobile coupons and makes the purchase decision based on the principle of maximum utility.

### 3.2. Gaming Sequence

Our analysis proceeds by examining the effect of mobile coupons in the context of a monopoly and a duopoly. In a monopoly, we consider a firm's optimal decisions on pricing and mobile coupon face value. In a duopoly, there are two firms competing in a two-stage game that seek to make optimal decisions on pricing and mobile coupon distribution.

The sequence of the duopoly game is as follows: In the initial stage, the firms choose whether to target consumers via mobile coupons. There are four subgames for the two firms as follows: (C, C), both firms adopt LBMC promotion; (C, NC) and (NC, C), only one firm adopts LBMC promotion; (NC, NC), no firm adopts LBMC promotion. Once the coupon strategy is set, the firms proceed to stage two to compete for consumers by simultaneously and non-cooperatively setting their retail prices and the face values of their mobile coupons. In stage three, all the consumers receive the mobile coupons and become aware of the prices and choose whether to buy the product from Firm 1 or Firm 2. Figure 1 depicts the sequence of the game events. We use subgame perfection as our solution concept.

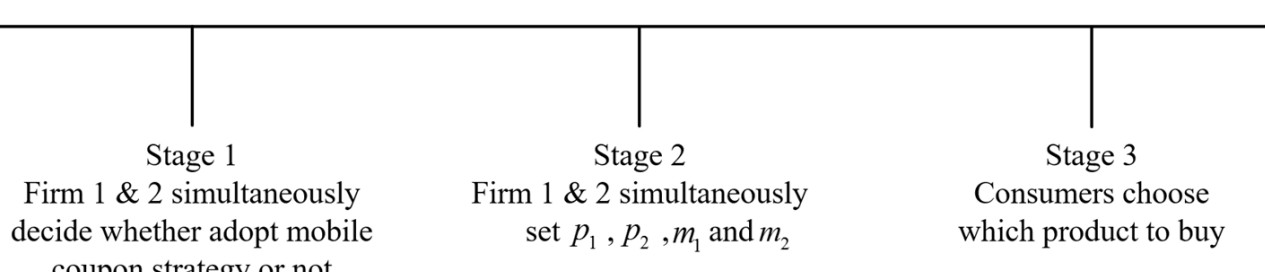

| Stage 1 | Stage 2 | Stage 3 |
|---|---|---|
| Firm 1 & 2 simultaneously decide whether adopt mobile coupon strategy or not | Firm 1 & 2 simultaneously set $p_1$, $p_2$, $m_1$ and $m_2$ | Consumers choose which product to buy |

**Figure 1.** Sequence of events.

## 4. Equilibrium Analysis

### 4.1. The Benchmarking Case: The Monopoly Model

We assume that a single firm (retailer) only occupies a part of the market, and it can push mobile coupons via mobile targeting technologies to areas in the residual market where consumers have non-positive utilities to compensate for the disutility of their temporal and spatial costs and lure the consumers in the competitor's area.

The consumers are distributed uniformly along a Hoteling linear city, where the firm is located at the left end of a unit line. The firm is to maximize its revenue, i.e., $px$. Thus, we obtain $x = \frac{V-p}{t}$, which is the indifferent point of the consumers. It is easy to derive that $p^* = \frac{V}{2}$, $x = \frac{V}{2t}$, $\pi^* = \frac{V^2}{4t}$, and $\frac{V}{t} \leq 2$.

Now the firm sends mobile coupons to target consumers on the right side of the indifferent point, i.e., $x = \frac{V-p}{t}$. Each consumer in this area receives a mobile coupon with a face value $m$ and will buy the product offered by the firm if he/she obtains a positive utility (see Figure 2).

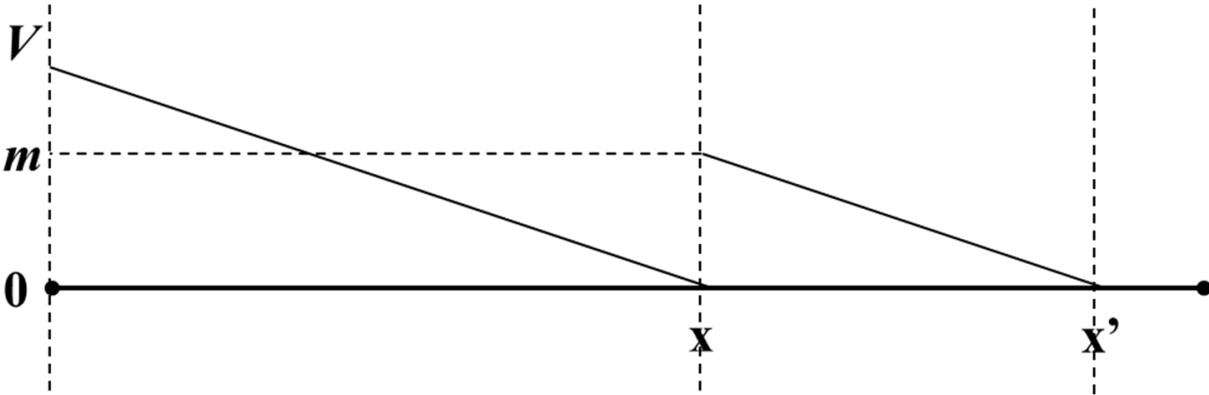

**Figure 2.** Mobile coupon targeting in the monopoly model.

**Lemma 1.** *In a monopoly market, the firm adopts LBMC promotion when the marginal targeting cost of mobile coupons satisfies $c < \frac{V}{2}$ and $t > \frac{2V-c}{3}$, i.e., the whole market is not fully covered. As a result, the total demand increases, while the non-targeted segment (the non-targeted segment is the area in which the retailer does not send mobile coupons to consumers) shrinks and the non-targeted consumers are charged a higher price. (Proof in Supplementary Materials).*

Lemma 1 reveals intuitively that the firm can target different prices to different consumers with mobile coupons based on consumer locations. Consumers in the non-targeted segment are charged a higher price and those in the targeted segment are compensated for their long-distance locations. As a result, the total sale increases in the form of the third degree of price discrimination. In other words, in the "blue ocean" market, the firm gains a larger market share and a higher profit. This result is consistent with existing theoretical models relating to when a firm should charge its old customers more than new customers. Shin and Sudhir [26] showed that when heterogeneity in purchase quantity and preference stochasticity are both low, it is optimal to reward the competitor's customers; however, the behavior-based pricing is less profitable in their models while LBMC promotion is more profitable in ours. This might be explained by the fact that firms can take advantage of the asymmetry of location information to price discriminate across different geographies.

### 4.2. Subgame (NC, NC) in the Duopoly Model

Consider the game (*NC, NC*), i.e., no firms use mobile coupons to target consumers. It is well known that the equilibrium outcomes in a horizontal differentiation model such as ours depend on the value of $\frac{V}{t}$. The ratio $\frac{V}{t}$ represents the cost-adjusted willingness of consumers to pay, much in the spirit of the notions of preference-adjusted valuation in Jerath et al. [61] and valuation-adjusted preference in Joshi et al. [62]. If $\frac{V}{t}$ is low, the degree of competition in the market is low and the firms act as local monopolies. As $\frac{V}{t}$ increases, the market covered increases until the firms begin to compete.

On the one hand, when $\frac{V}{t}$ is high, i.e., $\frac{V}{t} > \frac{3}{2}$, consumers have a relatively low switching cost, making the market inherently more competitive. Let $x_{NN}$ denote the location of the consumer's indifferent point between buying from Firm 1 and Firm 2, i.e., $V - p_1 - tx_{NN} = V - p_2 - t(1 - x_{NN})$, which yields $x_{NN} = \frac{p_2 - p_1 + t}{2t}$, and the firm profits are $\pi_1 = p_1 x_{NN}$ and $\pi_2 = p_2(1 - x_{NN})$. In equilibrium, the market is fully covered, and the optimal prices are $p_i^* = t$. The demand and profit of each firm is $\frac{1}{2}$ and $\frac{t}{2}$, respectively.

The consumer's indifferent point between buying from the two firms is at the center of the market and the consumers obtain a positive surplus.

On the other hand, when $\frac{V}{t}$ is low, i.e., $0 < \frac{V}{t} < 1$, consumers have a relatively high switching cost, making the market inherently less competitive. The market is not fully covered, and the equilibrium corresponds to the location of the consumer's indifferent point between purchasing firm $i$'s product or not, i.e., $\overline{x}_{NN}$. Thus, the equilibrium prices and demands are $p_i^* = \frac{V}{2}$ and $\frac{V}{2t}$, respectively, so each firm's profit is $\frac{V^2}{4t}$.

Finally, the intermediate values of $\frac{V}{t}$, i.e., $1 \le \frac{V}{t} \le \frac{3}{2}$, represent that the market is moderately competitive. In this case, there exist multiple price equilibria. Each of these equilibria corresponds to the location of a consumer's indifferent point between purchasing either firm's product $\hat{x}_{NN}$. Since the firms are *a priori* symmetric in the market, first, we focus only on the symmetric equilibrium (this is analogous to the focal point refinement [63], which is the most natural choice given the shared understanding of the environment by both firms) [62,64] such that the consumer's indifferent point between buying from either firm is $\hat{x}_{NN} = \frac{1}{2}$. For each $\hat{x}_{NN}$, the equilibrium prices are $p_1^* = V - t\hat{x}_{NN}$ and $p_2^* = V - t(1 - \hat{x}_{NN})$, the equilibrium demands are $d_1^* = \hat{x}_{NN}$ and $d_2^* = 1 - \hat{x}_{NN}$, and the equilibrium profits are $\pi_1^* = (V - t\hat{x}_{NN})\hat{x}_{NN}$ and $\pi_2^* = (V - t(1 - \hat{x}_{NN}))(1 - \hat{x}_{NN})$ for Firm 1 and Firm 2, respectively. Given the assumption, in this case, the equilibrium prices are $p_i^* = V - \frac{t}{2}$ and the equilibrium demands are $\frac{1}{2}$, so the equilibrium profit of each firm is $p_i^* = \frac{1}{2}(V - \frac{t}{2})$.

Table 2 summarizes the equilibria of subgame $(NC, NC)$.

**Table 2.** The equilibria of subgame $(NC, NC)$.

| $\frac{V}{t}$ | $p_i^*$ | $d_i^*$ | $\pi_i^*$ |
|---|---|---|---|
| $(0, 1)$ | $\frac{V}{2}$ | $\frac{V}{2t}$ | $\frac{V^2}{4t}$ |
| $[1, \frac{3}{2}]$ | $V - \frac{t}{2}$ | $\frac{1}{2}$ | $\frac{1}{2}(V - \frac{t}{2})$ |
| $(\frac{3}{2}, +\infty)$ | $t$ | $\frac{1}{2}$ | $\frac{t}{2}$ |

### 4.3. Subgames (C, NC) and (NC, C) in the Duopoly Model

In the subgame $(C, NC)$ or $(NC, C)$, only one firm uses mobile coupons to target consumers.

(1)   When the degree of competition is high (see Figure 3)

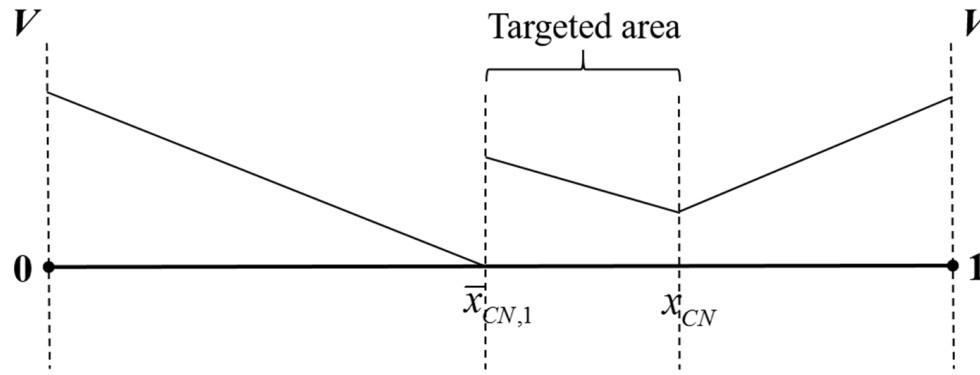

**Figure 3.** Subgame $(C, NC)$ under a high degree of competition.

The market is fully covered, and we obtain $V - p_1 - tx_{CN} + m_1 = V - p_2 - t(1 - x_{CN})$, $x_{CN} = \frac{p_2 - p_1 + m_1 + t}{2t}$, and $\overline{x}_{CN,1} = \frac{V - p_1}{t}$. Hence, the firms will maximize their profits as Equation (3):

$$\max \begin{cases} \pi_1 = p_1\overline{x}_{CN,1} + (p_1 - m_1 - c)(x_{CN} - \overline{x}_{CN,1}) \\ \pi_2 = p_2(1 - x_{CN}) \end{cases} \tag{3}$$

For $c < 3t - 2V$ and $\frac{V}{t} > \frac{6}{5}$, in equilibrium, $p_1^* = \frac{3t-V-c}{2}$, $m_1^* = \frac{3(V-t)-c}{2}$, and $p_2^* = 2t - V$. The total demands of the firms are $d_1 = \frac{V}{2t}$ and $d_2 = 1 - \frac{V}{2t}$, and the size of the targeted segment $T_1$ is $\frac{3}{2} - \frac{V}{t} - \frac{c}{2t}$. The equilibrium profits of the firms are $\pi_1 = \frac{(5V^2-12Vt+9t^2)+(c^2+(4V-6t)c)}{4t}$ and $\pi_2 = \frac{(2t-V)^2}{2t}$.

(2)  When the degree of competition is low (see Figure 4)

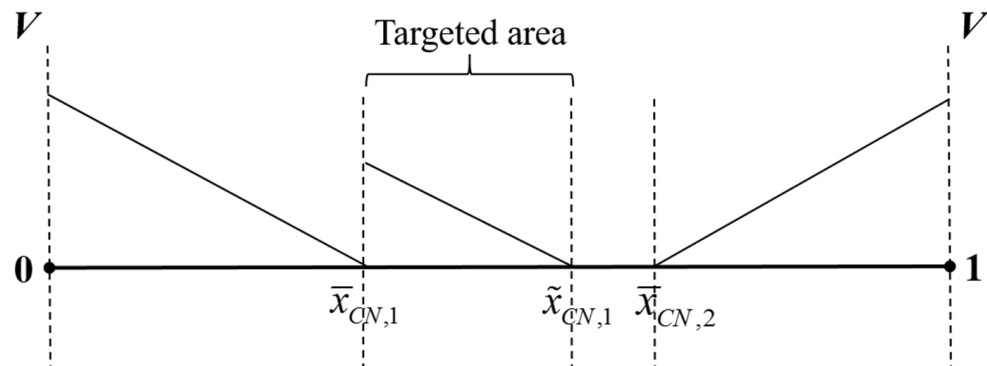

**Figure 4.** Subgame (*C*, *NC*) under a low degree of competition.

The whole market is not fully covered, and the firms will maximize

$$\begin{cases} \pi_1 = p_1 \bar{x}_{CN,1} + (p_1 - m_1 - c)(\tilde{x}_{CN,1} - \bar{x}_{CN,1}) \\ \pi_2 = p_2(1 - \bar{x}_{CN,2}) \end{cases} \tag{4}$$

where $\tilde{x}_{CN,1} = \frac{V-p_1+m_1}{t} < \bar{x}_{CN,2} = 1 - \frac{V-p_2}{t}$. For $0 < \frac{V}{t} < \frac{6}{7}$ and $c < \frac{V}{2}$, the equilibrium outcomes are $p_1^* = \frac{2V-c}{3}$, $m_1^* = \frac{V-2c}{3}$, and $p_2^* = \frac{V}{2}$. The total demands of the firms are $d_1 = \frac{2V-c}{3t}$ and $d_2 = \frac{V}{2t}$, and the size of the targeted segment $T_1$ is $\frac{V}{3t}$. The equilibrium profits of the firms are $\pi_1 = \frac{V^2-Vc+c^2}{3t}$ and $\pi_2 = \frac{V^2}{4t}$.

(3)  When the degree of competition is moderate (see Figure 5)

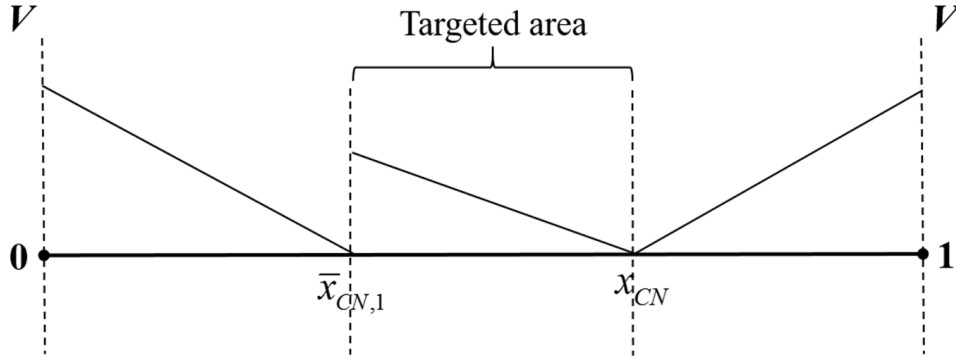

**Figure 5.** Subgame (*C*, *NC*) under a moderate degree of competition.

As in Section 4.2, for $\frac{6}{7} \le \frac{V}{t} \le \frac{6}{5}$ and $c < \frac{t}{2}$, the consumer's indifferent point between buying from either firm is $\hat{x}_{CN} = \frac{1}{2}$. The equilibrium outcomes are $p_1^* = V - \frac{t}{4} - \frac{c}{2}$, $m_1^* = \frac{t}{4} - \frac{c}{2}$, and $p_2^* = V - \frac{t}{2}$. The total demands of the firms are $d_i = \frac{1}{2}$ and the size of the targeted segment $T_1$ is $\frac{1}{4} - \frac{c}{2t}$. The equilibrium profits of the firms are $\pi_1 = \frac{V}{2} - \frac{3t}{16} + \frac{c^2-tc}{4t}$ and $\pi_2 = \frac{V}{2} - \frac{t}{4}$.

Table 3 summarizes the equilibria of subgame (*C*, *NC*).

**Table 3.** The equilibria of subgame (*C*, *NC*).

| $\frac{V}{t}$ | $(0,\frac{6}{7})$ | $[\frac{6}{7},\frac{6}{5}]$ | $(\frac{6}{5},\frac{3}{2})$ |
|---|---|---|---|
| $c$ | $(0,\frac{V}{2})$ | $(0,\frac{t}{2})$ | $(0,3t-2V)$ |
| $p_1^*$ | $\frac{2V-c}{3}$ | $V-\frac{t}{4}-\frac{c}{2}$ | $\frac{3t-V-c}{2}$ |
| $m_1^*$ | $\frac{V-2c}{3}$ | $\frac{t}{4}-\frac{c}{2}$ | $\frac{3(V-t)-c}{2}$ |
| $p_2^*$ | $\frac{V}{2}$ | $V-\frac{t}{2}$ | $2t-V$ |
| $d_1$ | $\frac{2V-c}{3t}$ | $\frac{1}{2}$ | $\frac{V}{2t}$ |
| $T_1$ | $\frac{V}{3t}$ | $\frac{1}{4}-\frac{c}{2t}$ | $\frac{3}{2}-\frac{2V+c}{2t}$ |
| $d_2$ | $\frac{V}{2t}$ | $\frac{1}{2}$ | $1-\frac{V}{2t}$ |
| $\pi_1$ | $\frac{V^2-Vc+c^2}{3t}$ | $\frac{V}{2}-\frac{3t}{16}+\frac{c^2-tc}{4t}$ | $\frac{(5V^2-12Vt+9t^2)+(c^2+(4V-6t)c)}{4t}$ |
| $\pi_2$ | $\frac{V^2}{4t}$ | $\frac{V}{2}-\frac{t}{4}$ | $\frac{(2t-V)^2}{2t}$ |

**Lemma 2.** *In the asymmetric subgame (C, NC) or (NC, C), the firm that adopts LBMC promotion is always better off, while the other firm is no better off. Moreover, when the degree of competition of the market is relatively high, i.e., $\frac{V}{t} \in (\frac{6}{5},\frac{3}{2})$, the other firm is worse off. (Proof in Supplementary Materials).*

Lemma 2 reveals that both firms have incentives to adopt LBMC promotion to obtain higher profits. The non-adopter is not affected when the competition intensity is not high; but it is not the case when the competition intensity is relatively high. Thus, the subgames (*NC*, *C*) and (*C*, *NC*) do not have stable equilibria.

**Corollary 1.** *Unilateral adoption of LBMC promotion intensifies the competition in the market. (Proof in Supplementary Materials).*

*4.4. Subgame (C, C) in the Duopoly Model*

(1)  When the competition degree is high

The market is fully covered, and we have $V-p_1-tx_{CC}+m_1=V-p_2-t(1-x_{CC})+m_2$, $x_{CC}=\frac{p_2-p_1+m_1-m_2+t}{2t}$, and $\overline{x}_{CC,1}=\frac{V-p_1}{t}$ $\overline{x}_{CC,2}=1-\frac{V-p_2}{t}$. The firms will maximize their profits are in Equation (5):

$$\max\begin{cases} \pi_1 = p_1\overline{x}_{CC,1}+(p_1-m_1-c)(x_{CC}-\overline{x}_{CC,1}) \\ \pi_2 = p_2(1-\overline{x}_{CC,2})+(p_2-m_2-c)(\overline{x}_{CC,2}-x_{CC}) \end{cases} \tag{5}$$

In equilibrium, the targeted area is zero, which implies that both firms choose not to use mobile coupons to target consumers.

(2)  When the degree of competition is low

The whole market is not fully covered, and the firms' profits are in Equation (6):

$$\max\begin{cases} \pi_1 = p_1\overline{x}_{CC,1}+(p_1-m_1-c)\frac{m_1}{t} \\ \pi_2 = p_2(1-\overline{x}_{CC,2})+(p_2-m_2-c)\frac{m_2}{t} \end{cases} \tag{6}$$

where $\widetilde{x}_{CC,1}=\frac{V-p_1+m_1}{t}<\widetilde{x}_{CC,1}=1-\frac{V-p_2+m_2}{t}$. For $0<\frac{V}{t}<\frac{3}{2}$ and $c<\frac{V}{2}$, the equilibrium outcomes are $p_i^*=\frac{2V-c}{3}$ and $m_i^*=\frac{V-2c}{3}$. The total demands of the firms are $d_i=\frac{2V-c}{3t}$ and the size of the targeted segment $T_i$ is $\frac{V-2c}{3t}$. The equilibrium profit of each firm is $\pi_1=\frac{V^2-c^2}{3t}$.

(3)  When the competition degree is moderate

Since both firms choose not to adopt LBMC promotion, if $\frac{V}{t}$ is relatively high, the equilibrium corresponds to the location of the consumer's indifferent point between purchasing

either firm's product $\hat{x}_{CC} = \frac{1}{2}$. For $\frac{V}{t} \geq \frac{3}{4}$, the equilibrium outcomes are $p_1^* = V - \frac{t}{4} - \frac{c}{2}$ and $m_1^* = \frac{t}{4} - \frac{c}{2}$, so the equilibrium profit of each firm is $\pi_i = \frac{V}{2} - \frac{3t}{16} + \frac{c^2 - tc}{4t}$.

Table 4 summarizes the equilibria of subgame $(C, C)$.

**Table 4.** The equilibria of subgame $(C, C)$.

| $\frac{V}{t}$ | $c$ | $p_i^*$ | $m_i^*$ | $d_i^*$ | $T_i^*$ | $\pi_i^*$ |
|---|---|---|---|---|---|---|
| $(0, \frac{3}{4})$ | $(0, \frac{V}{2})$ | $\frac{2V-c}{3}$ | $\frac{V-2c}{3}$ | $\frac{2V-c}{3t}$ | $\frac{V-2c}{3t}$ | $\frac{V^2-c^2}{3t}$ |
| $[\frac{3}{4}, +\infty)$ | $(0, \frac{t}{2})$ | $V - \frac{t}{4} - \frac{c}{2}$ | $\frac{t}{4} - \frac{c}{2}$ | $\frac{1}{2}$ | $\frac{1}{2}$ | $\frac{V}{2} - \frac{3t}{16} + \frac{c^2-tc}{4t}$ |

**Lemma 3.** *In the symmetric subgame $(C, C)$, both firms are better off as compared with the subgame when no firm adopts LBMC promotion, i.e., subgame $(NC, NC)$. In addition, both subgames $(NC, C)$ and $(C, NC)$ have no stable equilibria and will deviate from subgame $(C, C)$ ultimately. (Proof in Supplementary Materials).*

*4.5. LBMC Promotion in the Duopoly Market*

Comparing the profits among the various subgames, we obtain after tedious derivations the SPNE as follows:

**Proposition 1.** *In the competitive duopoly market, there is a unique SPNE at which both firms adopt LBMC promotion under all the scenarios if the conditions in Table 5 are satisfied. Otherwise, no firm adopts LBMC promotion in the rest of the region. (Proof in Supplementary Materials).*

**Table 5.** The pure strategy equilibria in mobile coupon decisions.

| SPNE | $\frac{V}{t}$ | $c$ |
|---|---|---|
| $(C, C)$ | $(0, \frac{3}{4})$<br>$[\frac{3}{4}, \frac{5}{4}]$<br>$(\frac{5}{4}, \frac{3}{2})$<br>$[\frac{3}{2}, +\infty)$ | $(0, \frac{V}{2})$<br>$(0, \frac{t}{2})$<br>$(0, 3t - 2V)$<br>$(0, \frac{t}{2})$ |

We apply $\frac{V}{t}$ to partition the market under different competition intensities to derive the cost conditions for adopting LBMC promotion in equilibrium. Evidently, Proposition 1 implies that the marginal cost of using mobile coupons is a crucial determinant for firms to adopt LBMC promotion, i.e., only when the marginal targeting cost is below the threshold. If the marginal targeting cost exceeds the threshold, both firms will not adopt LBMC promotion. Changes in consumer's willingness to pay and travel cost have different impacts on the marginal cost. For instance, when $\frac{V}{t} \in (0, \frac{3}{4})$, the threshold of the marginal cost $\frac{V}{2}$ remains constant as the travel cost $t$ changes. When $\frac{V}{t} \in \{[\frac{3}{4}, \frac{5}{4}] \cup [\frac{3}{2}, +\infty)\}$, the threshold of the marginal cost $\frac{t}{2}$ remains constant as consumer's willingness to pay $V$ changes. When $\frac{V}{t} \in (\frac{5}{4}, \frac{3}{2})$, consumer's willingness to pay and travel cost have joint effects on the threshold of the marginal cost $3t - 2V$. These results provide useful insights about the marginal targeting cost of mobile coupons. As stated above, if the marginal targeting cost of mobile coupons exceeds a threshold, no firm chooses LBMC promotion. This implies that coupon publishers should strengthen their R&D on location-based services to lower the targeting cost, which draws more firms to adopt LBMC promotion. Therefore, this will result in a win-win situation (it implies that both firms are better off as compared with no couponing) for both coupon publishers and retailers.

Proposition 1 provides important managerial implications for marketing practitioners seeking to formulate effective strategies for targeted LBMC promotion. It indicates that mobile coupons are an effective means for retailers to enhance their power to entice consumers. It is important to realize that contemporary retailing is increasingly becoming LBS oriented that makes use of the bricks-and-mortar and mobile channels. Brick-and-mortar retailers should employ new tools to attract more consumers in the omni-channel

era. As a new promotion tool, LBMCs have the advantages of segmenting consumers based on their real-time locations and generating more consumer traffic to retail stores. Brick-and-mortar retailers should opt for the proper strategy for adopting LBMC promotion.

Our findings shed light on a retailer's decisions on the adoption of LBMC promotion in a competitive market. The adoption of LBMC promotion can yield a competitive edge to a retailer over a conventional firm because the adopter firm has a greater market reach (when the degree of competition is low) and is able to cater to consumers in different locations. It captures the consumer surplus with geographical privilege (i.e., consumers in the vicinity of a physical store), which compensates consumers in a faraway place with the third degree of price discrimination. Hence, a single firm can benefit from adopting targeted LBMC promotion, while the competing firm also has the incentive to adopt mobile coupons. For LBMC, asymmetric adoption does not exist if the marginal targeting cost is low enough and the optimal equilibrium results in win-win outcomes. In the presence of a competitor, the retailer should adopt LBMC promotion if the marginal targeting cost is low enough.

## 5. Extension

In this section, we extend our model by considering the competing firms may differ in the quality of their products. Specifically, consumers are heterogeneous in their willingness to pay for the goods of different quality. We assume that the reservation value for the product from firm $i$ is $V_i$, which depends on the product quality of the firm [65]. Without loss of generality, let Firm 1 be the superior-quality firm with its product quality normalized to one, i.e., $V_1 = V$. Firm 2 is the inferior-quality firm with the quality of its product $q \leq 1$, i.e., $V_2 = qV$. Consumer $l$'s utility from buying the product from firm $i$ without mobile coupons is in Equation (7):

$$U_l = \begin{cases} V - p_1 - tx_l \\ qV - p_2 - t(1 - x_l) \end{cases}, without \ mobile \ coupons \qquad (7)$$

Consumer $l$'s utility from buying the product from firm $i$ with mobile coupons is in Equation (8):

$$U_l = \begin{cases} V - p_1 - tx_l + m_1 \\ qV - p_2 - t(1 - x_l) + m_2 \end{cases}, with \ mobile \ coupons \qquad (8)$$

Similar to the analysis in Section 4, we can derive the equilibrium results of the four subgames (*NC, NC*), (*C, NC*), (*NC, C*), and (*C, C*) under the condition of quality differentiation (see Tables 6–9).

**Table 6.** The equilibria of subgame (*NC, NC*) under the condition of quality differentiation.

| $\frac{V}{t}$ | $(0, \frac{2}{1+q})$ | $[\frac{2}{1+q}, \frac{3}{1+q}]$ | $(\frac{3}{1+q}, +\infty)$ |
|---|---|---|---|
| $p_1^*$ | $\frac{V}{2}$ | $V - \frac{t}{2}$ | $t + \frac{(1-q)}{3}V$ |
| $p_2^*$ | $\frac{qV}{2}$ | $qV - \frac{t}{2}$ | $t - \frac{(1-q)}{3}V$ |
| $d_1$ | $\frac{V}{2t}$ | $\frac{1}{2}$ | $\frac{1}{2} + \frac{(1-q)V}{6t}$ |
| $d_2$ | $\frac{qV}{2t}$ | $\frac{1}{2}$ | $\frac{1}{2} - \frac{(1-q)V}{6t}$ |
| $\pi_1$ | $\frac{V^2}{4t}$ | $\frac{V}{2} - \frac{t}{4}$ | $\frac{[3t+(1-q)V]^2}{18t}$ |
| $\pi_2$ | $\frac{q^2V^2}{4t}$ | $\frac{qV}{2} - \frac{t}{4}$ | $\frac{[3t-(1-q)V]^2}{18t}$ |

**Table 7.** The equilibria of subgame (C, NC) under the condition of quality differentiation.

| $\frac{V}{t}$ | $(0, \frac{6}{4+3q})$ | $[\frac{6}{4+3q}, \frac{6}{3+2q}]$ | $(\frac{6}{3+2q}, \frac{3}{1+q})$ |
|---|---|---|---|
| $c$ | $(0, \frac{V}{2})$ | $(0, \frac{t}{2})$ | $(0, 3t-(1+q)V)$ |
| $p_1^*$ | $\frac{2V-c}{3}$ | $V - \frac{t}{4} - \frac{c}{2}$ | $\frac{3t-qV-c}{2}$ |
| $m_1^*$ | $\frac{V-2c}{3}$ | $\frac{t}{4} - \frac{c}{2}$ | $\frac{(2+q)V-3t-c}{2}$ |
| $p_2^*$ | $\frac{qV}{2}$ | $qV - \frac{t}{2}$ | $2t - V$ |
| $d_1$ | $\frac{2V-c}{3t}$ | $\frac{1}{2}$ | $\frac{V}{2t}$ |
| $T_1$ | $\frac{V}{3t}$ | $\frac{1}{4} - \frac{c}{2t}$ | $\frac{3}{2} - \frac{(1+q)V+c}{2t}$ |
| $d_2$ | $\frac{V-2c}{3t}$ | $\frac{1}{2}$ | $1 - \frac{V}{2t}$ |
| $\pi_1$ | $\frac{V^2-Vc+c^2}{3t}$ | $\frac{V}{2} - \frac{3t}{16} + \frac{c^2-tc}{4t}$ | $\frac{[(q^2+2q+2)V^2-6(1+q)Vt+9t^2)]+c^2+2(1+q)Vc-6tc}{4t}$ |
| $\pi_2$ | $\frac{q^2V^2}{4t}$ | $\frac{qV}{2} - \frac{t}{4}$ | $\frac{(2t-V)^2}{2t}$ |

**Table 8.** The equilibria of subgame (NC, C) under the condition of quality differentiation.

| $\frac{V}{t}$ | $(0, \frac{6}{3+4q})$ | $[\frac{6}{3+4q}, \frac{6}{2+3q}]$ | $(\frac{6}{2+3q}, \frac{3}{1+q})$ |
|---|---|---|---|
| $c$ | $(0, \frac{V}{2})$ | $(0, \frac{t}{2})$ | $(0, 3t-(1+q)V)$ |
| $p_1^*$ | $\frac{V}{2}$ | $V - \frac{t}{2}$ | $2t - qV$ |
| $p_2^*$ | $\frac{2qV-c}{3}$ | $qV - \frac{t}{4} - \frac{c}{2}$ | $\frac{3t-V-c}{2}$ |
| $m_2^*$ | $\frac{qV-2c}{3}$ | $\frac{t}{4} - \frac{c}{2}$ | $\frac{(1+2q)V-3t-c}{2}$ |
| $d_1$ | $\frac{qV}{2t}$ | $\frac{1}{2}$ | $1 - \frac{qV}{2t}$ |
| $d_2$ | $\frac{2V-c}{3t}$ | $\frac{1}{2}$ | $\frac{qV}{2t}$ |
| $T_2$ | $\frac{qV-2c}{3t}$ | $\frac{1}{4} - \frac{c}{2t}$ | $\frac{3}{2} - \frac{(1+q)V+c}{2t}$ |
| $\pi_1$ | $\frac{V^2}{4t}$ | $\frac{V}{2} - \frac{t}{4}$ | $\frac{(2t-V)^2}{2t}$ |
| $\pi_2$ | $\frac{q^2V^2-qVc+c^2}{3t}$ | $\frac{qV}{2} - \frac{3t}{16} + \frac{c^2-tc}{4t}$ | $\frac{[(2q^2+2q+1)V^2-6(1+q)Vt+9t^2)]+c^2+2(1+q)Vc-6tc}{4t}$ |

**Table 9.** The equilibria of subgame (C, C) under the condition of quality differentiation.

| $\frac{V}{t}$ | $(0, \frac{3}{2(1+q)})$ | $[\frac{3}{2(1+q)}, +\infty)$ |
|---|---|---|
| $c$ | $(0, \frac{V}{2})$ | $(0, \frac{t}{2})$ |
| $p_1^*$ | $\frac{2V-c}{3}$ | $V - \frac{t}{4} - \frac{c}{2}$ |
| $m_1^*$ | $\frac{V-2c}{3}$ | $\frac{t}{4} - \frac{c}{2}$ |
| $p_2^*$ | $\frac{2qV-c}{3}$ | $qV - \frac{t}{4} - \frac{c}{2}$ |
| $m_2^*$ | $\frac{qV-2c}{3}$ | $\frac{t}{4} - \frac{c}{2}$ |
| $d_1$ | $\frac{2V-c}{3t}$ | $\frac{1}{2}$ |
| $T_1$ | $\frac{V-2c}{3t}$ | $\frac{1}{4} - \frac{c}{2t}$ |
| $d_2$ | $\frac{V-2c}{3t}$ | $\frac{1}{2}$ |
| $T_2$ | $\frac{qV-2c}{3t}$ | $\frac{1}{4} - \frac{c}{2t}$ |
| $\pi_1$ | $\frac{V^2-Vc+c^2}{3t}$ | $\frac{V}{2} - \frac{3t}{16} + \frac{c^2-tc}{4t}$ |
| $\pi_2$ | $\frac{q^2V^2-qVc+c^2}{3t}$ | $\frac{qV}{2} - \frac{3t}{16} + \frac{c^2-tc}{4t}$ |

We performed numerical studies to illustrate firms' choices of strategies for LCMC promotion with quality differentiation by comparing their equilibrium profits among the subgames under different market competition intensities. We make the following observations from the numerical studies:

(1) In a market with a high degree of competition, i.e., $\frac{V}{t} > \frac{3}{1+q}$, and the marginal targeting cost is $c < \frac{t}{2}$, we assume $V = 1$, $t = 0.2$, and $c < 0.15$ (for the plots, the values of the parameters $V$ and $t$ follow those used in [61], and the marginal targeting cost $c$ is determined by $V$ or $t$), and classify Firm 2's quality level as low ($q = 0.1$), medium ($q = 0.5$), and high ($q = 0.9$). We show the results in Figure 6.

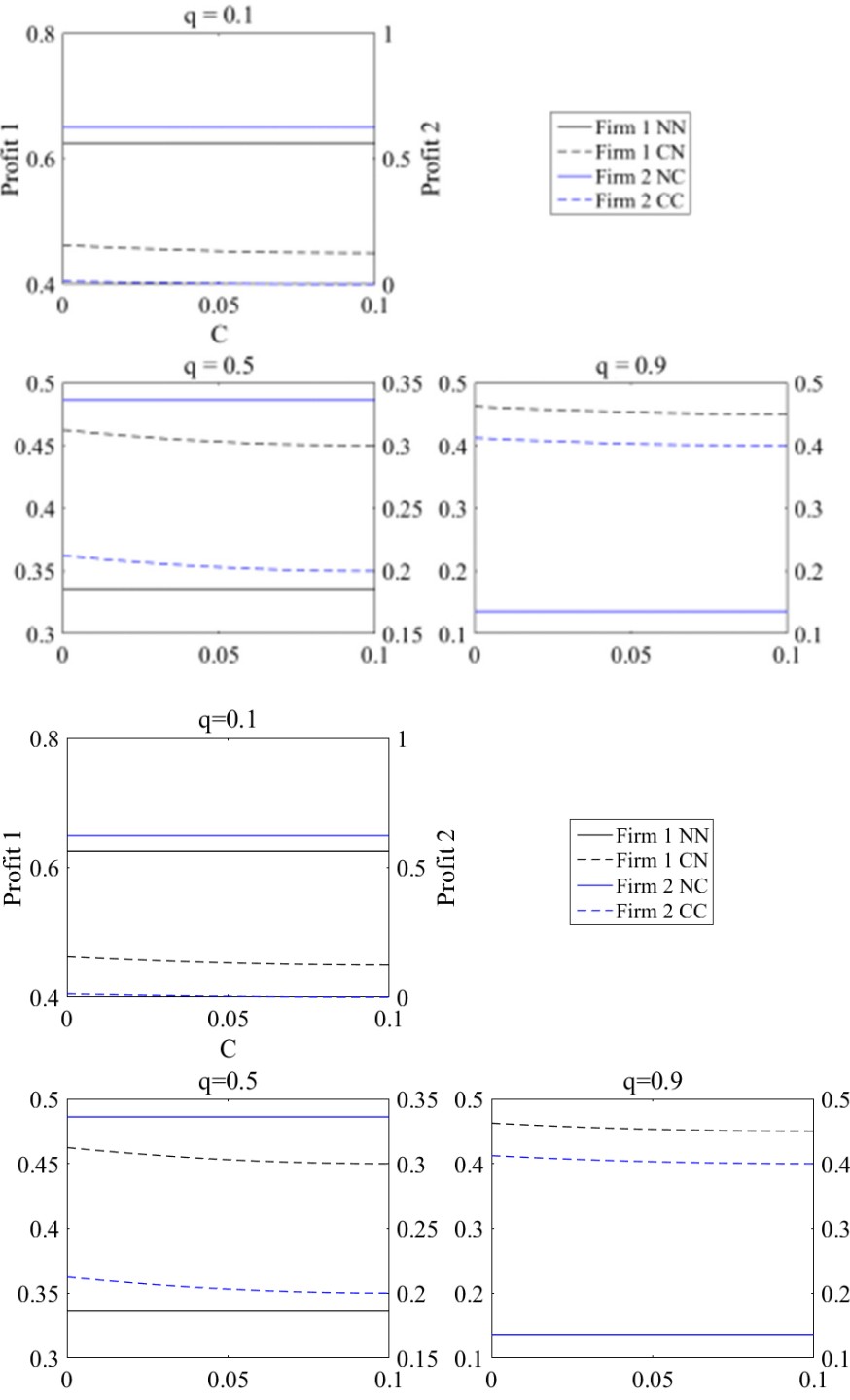

**Figure 6.** Firm profits of subgames under a high degree of competition.

If quality differentiation is low, i.e., *q* is high, both firms adopt LBMC promotion, which is similar to the conclusions in Section 4. However, the results change when quality differentiation increases. If quality differentiation is medium, there is a mixed Nash equilibrium at which the inferior-quality firm does not adopt LBMC promotion. With quality differentiation further increasing to a high level, no firm adopts LBMC promotion eventually. In reality, if quality differentiation is high, the firm with superior product quality will exploit its quality advantage to compete for consumers and will not adopt LBMC promotion. Its quality advantage is enough for it to compete with rivals with inferior product quality. For instance, in the fast-food industry (Domino's, Pizza Hut, and Burger

King) LBS technology is applied to locate smartphone users and targeted to reach potential customers who are near their stores.

(2)    In a market with a medium degree of competition, i.e., $\frac{3}{2(1+q)} < \frac{V}{t} < \frac{3}{1+q}$, and the marginal targeting cost is $c < \frac{t}{2}$, we assume $V = 1$, $t = 0.8$, and $c < 0.4$, and classify Firm 2′s quality level as low ($q = 0.1$), medium ($q = 0.5$), and high ($q = 0.9$). We show the results in Figure 7.

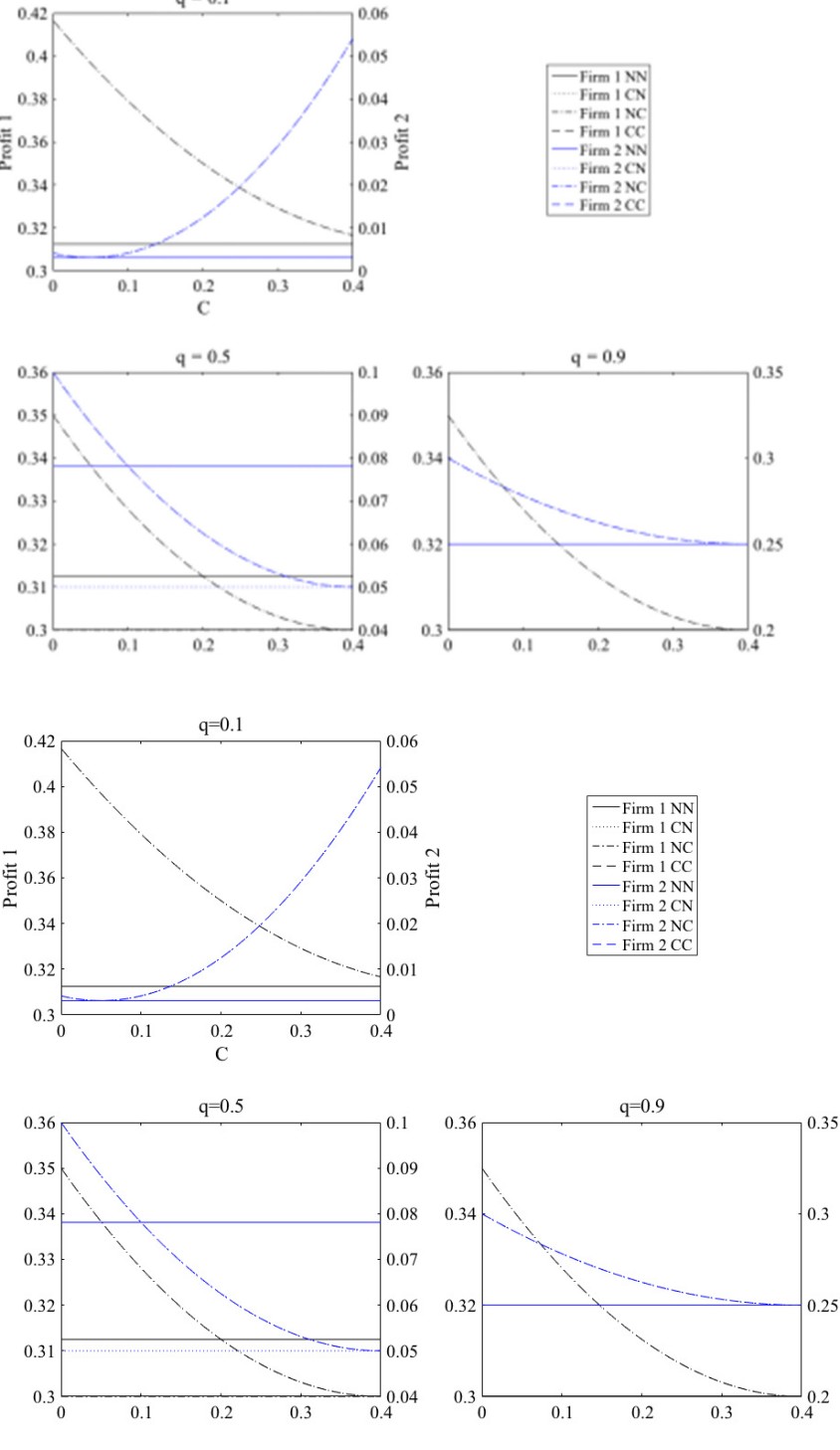

**Figure 7.** Firm profits of subgames under a medium degree of competition.

When quality differentiation is low or high, i.e., $q = 0.1/0.9$, the adoption of LBMC promotion is always an optimal strategy for both firms. When quality differentiation is

medium, the superior-quality firm's choice remains unchanged while the inferior-quality firm's optimal choice changes to not adopting LBMC promotion if the marginal targeting cost is relatively high, i.e., $c > 0.11$.

(3) In a market with a low degree of competition, i.e., $\frac{V}{t} < \frac{3}{2(1+q)}$, and the marginal targeting cost is $c < \frac{V}{2}$, we assume $V = 1$, $t = 2$, and $c < 0.5$, and classify Firm 2's quality level as low ($q = 0.1$), medium ($q = 0.5$), and high ($q = 0.9$). We show the results in Figure 8.

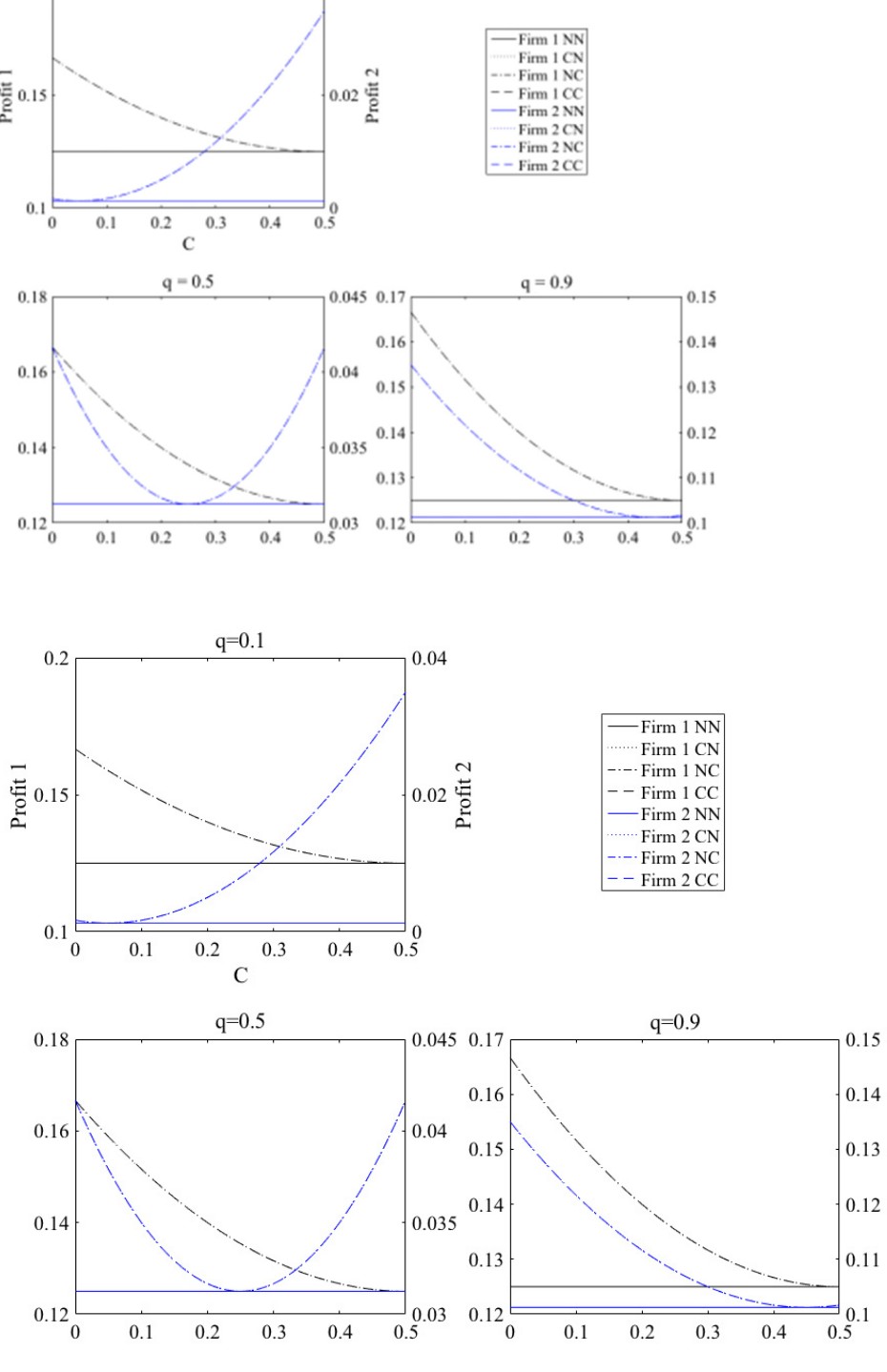

**Figure 8.** Firm profits of subgames under a low degree of competition.

Under this scenario, quality differentiation has no impact on firms' choices of strategies and the adoption of LBMC promotion is always the optimal strategy for both firms because their markets are independent of each other. This result is in line with the findings in Proposition 1.

In summary, quality differentiation has a significant impact on the optimal strategy choice of the inferior-quality firm, while the superior-quality firm's choice of strategy changes only when quality differentiation is very high under a high degree of competition. In Proposition 1, firms have only symmetric optimal pure strategies (i.e., no adoption and symmetric adoption) as long as the marginal targeting cost of mobile coupons $c$ satisfies certain conditions. However, mixed strategies for LBMC promotion emerge when firms' products have different quality. For instance, when quality differentiation is moderate and the degree of market competition is medium or high, the pure strategy does not exist.

**Corollary 2.** *The superior-quality firm that has the quality advantage always makes a higher (or the same) profit in competing with the inferior-quality firm under the scenario of symmetric adoption or no adoption of LBMC promotion.*

Furthermore, comparing the equilibrium profits of the firms under the scenario of symmetric adoption or no adoption of LBMC promotion, we find that the superior-quality firm's equilibrium profit is higher than those under all other scenarios. It is evident that the superior-quality firm has the quality advantage when both firms adopt LBMC promotion. As the whole market becomes saturated, i.e., the degree of competition is medium or even high, LBMC promotion cannot attract more consumers. Thus, when both firms incur the same level of marketing expenses, the superior-quality firm achieves better results due to higher consumer valuations for purchasing the higher quality product that it sells.

As we know that, under symmetric competition, a retailer should adopt LBMC promotion as long as the marginal targeting cost is low enough. However, this is no longer the case in the asymmetric competition environment. If quality differentiation is extremely low or high, adopting LBMC promotion is the optimal strategy (except under the scenario of high competition and high-quality differentiation, in which case no adoption of LBMC promotion is the optimal strategy). When quality differentiation is medium, firms' profits are determined by the degree of competition, quality differentiation, and marginal targeting cost of mobile coupons, in which case mixed strategies exist. What counts for the superior-quality firm is that with its quality advantage, it always makes a higher profit from competing with LBMC promotion.

## 6. Conclusions
### 6.1. Discussion of the Main Findings

As increasing attention—both academic and practical—has been paid to mobile coupons, LBMC promotion has brought new marketing problems for retailers to address. The central issue facing a retailer is to formulate effective strategies for LBMC promotion to discriminate customers in different regions and compete with rival firms. In this paper, we present a game model to analyze the optimal strategies for LBMC promotion for two firms competing in a market with a horizontally differentiated product. We derive the optimal strategies for the two firms under different conditions of the marginal targeting cost of mobile coupons and market competition intensities. Moreover, we extend the model to consider quality differentiation between the two firms and illustrate how the optimal strategies for LBMC promotion change with quality differentiation.

We make three contributions in this research: (i) We demonstrate that mobile coupons are an effective marketing tool that can compensate consumers for their disutility in the purchasing process. As compared with mass promotion, mobile coupons can take advantage of consumers' various locations, which allows firms to exercise price discrimination to increase the consumption demand. Previous studies have found that the effectiveness of mobile coupons depended on contextual variables and coupon characteristics [11,57].

In contrast with previous studies, we focus on analyzing firms' strategies for adopting LBMC promotion, whereby firms choose to compete for consumers located in the remote segment for which mobile coupons play the important role of price discrimination. (ii) We derive the conditions under which firms should adopt LBMC promotion under different market competition intensities and at different targeting costs. Previous research has not addressed the question of how to compete for location-targeted customers by taking advantage of LBSs and mobile coupons, despite the fact that, increasingly, firms are adopting this promotion strategy. Moreover, there has been no empirical and theoretical analyses of the ideal competitive response to a competitor's adoption of mobile promotion [6,19,66]. (iii) We find that quality differentiation and the marginal targeting cost of mobile coupons have joint impacts on firms. The superior-quality firm that has the quality advantage always makes higher (or the same) profits in competing with the inferior-quality firm in the symmetric competition environment. Our findings are consistent with the phenomena often observed in service systems such as restaurants and retail stores, in which firms with better targeting and integration of multiple channels in their marketing plans, such as Sears and Kmart, are making LBMCs a more intricate part of their promotion mixes.

In this paper, we provide useful insights for managers and marketing practitioners seeking to formulate effective strategies for targeted LBMC promotion. Our findings indicate that mobile coupons are an effective means for retailers to enhance their power to entice consumers. It is important to realize that contemporary retailing is increasingly becoming LBS oriented, which makes use of the bricks-and-mortar and mobile channels. Brick-and-mortar retailers should employ new tools to attract more consumers in the omni-channel era. As a new promotion tool, LBMCs have the advantages of segmenting consumers based on their real-time locations and generating more consumer traffic to retail stores. Brick-and-mortar retailers should opt for the proper strategy for adopting LBMC promotion.

Our findings shed light on a retailer's decisions on the adoption of LBMC promotion in a competitive market. The adoption of LBMC promotion can yield a competitive edge to a retailer over a conventional firm, which reveals that retailers should turn to LBMCs to drive consumer traffic to their retail stores. In the presence of a competitor, the retailer should adopt LBMC promotion as long as the marginal targeting cost is low enough. However, this result depends on the degree of quality differentiation in the asymmetric competition environment. We also find that a win-win situation for both coupon publishers and retailers emerge if the former enhance their R&D on location-based services to lower the targeting cost.

*6.2. Limitations and Opportunities for Future Research*

This study as several limitations that we suggest as directions for future research. First, the combination of consumers' preferences and location information can make targeted LBMC promotion more precise, while the cost of consumer recognition should be taken into consideration. In addition, our model does not cover all of the complex aspects of the real market, for example, the characteristics of the temporal effect [6] and geo-conquesting [66] are not considered. We do, however, provide a theoretical analysis to account for the increasing adoption of LBMC promotion. As our theoretical findings provide empirically testable hypotheses, future research should conduct such empirical studies.

**Supplementary Materials:** The following are available online at https://www.mdpi.com/article/10.3390/jtaer16070176/s1, Proof of Lemma 1, Proof of Lemma 2, Proof of Corollary 1, Proof of Lemma 3 and Proof of Proposition 1.

**Author Contributions:** Conceptualization, G.L. and P.X.; methodology, P.X.; software, P.X.; validation, T.C.E.C., G.L. and P.X.; formal analysis, P.X.; writing—original draft preparation, P.X.; writing—review and editing, T.C.E.C., G.L., P.X. and A.S.; supervision, T.C.E.C. and G.L.; project administration, T.C.E.C. and G.L.; funding acquisition, G.L. All authors have read and agreed to the published version of the manuscript.

**Funding:** This research is supported by National Natural Science Foundation of China "A study of operations management and resource configuration in sharing economy environment, 71832011". This research is also supported by Shaanxi Science and Technology Innovation Team Plan "Innovation team of service-oriented and intelligent manufacturing management of high-end equipment based on sharing economy, S2020-ZC-TD-0083".

**Institutional Review Board Statement:** Not applicable.

**Informed Consent Statement:** Not applicable.

**Data Availability Statement:** Not applicable.

**Conflicts of Interest:** We declare that we have no financial and personal relationships with other people or organizations that can inappropriately influence our work, there is no professional or other personal interest of any nature or kind in any product, service and/or company that could be construed as influencing the position presented in, or the review of, the manuscript entitled.

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
