# Peer review of "Competition Strategies for Location-Based Mobile Coupon Promotion"

_jtaer, doi:10.3390/jtaer16070176_

Round 1

Reviewer 1 Report

Dear author/authors,

The article addresses an interesting topic and is well structured, but some corrections and improvements are needed, as follows:

  • within the Abstract there is a specification regarding the analysis of the implications of adopting LBMC promotion for the superior-quality and inferior-quality firms, but it is necessary to clarify what the inferior and superior quality of a company represents, to what exactly you refer;
  • the Introduction section is much too extensive and part of it can rather be considered a review of the literature;
  • a Discussions section is required, and this part should focus on presenting implication and correlating the research results with other studies;
  • bibliography contains many obsolete sources of documentation and relates very little to new, current bibliographic resources, necessary when discussing such a current topic.

Reviewer 2 Report

The subject of the paper “ Competition Strategies for Location-based Mobile Coupon Promotion” is timely and valuable to the audience of the JTAER. Researchers presented a model of two competing retailers in which each retailer can send mobile coupons via mobile targeting technologies to areas in the residual market to attract the consumers.

Overall, the paper is well structured, reads quite well, and covers the existing literature quite well. The analysis of the data is interesting and well documented. However, to my view, some minor amendments are required prior to publication.

The only thing is to better include references into text and section references needs strong support. The list of references is not well presented. MPDI use own citation style with abbreviated journal names.  There are some missing details, like e.g. lack of DOI numbers and lack of page numbers. MPDI has own citation style, you should use it. The same applies to in-text citations. Citing several papers in document needs to use dash character or comma. 

Round 2

Reviewer 1 Report

Dear author/authors,

Given that you have considered the suggestions in the previous review report, I appreciate that the article is now ready for publication.

Reviewer 2 Report

Thank you very much. All of my previous comments were correctly addressed. Thank you very much for clarifying the references. I think that the manuscript has been significantly improved. I wish you good luck in your future work.